# Resilience of Canola to *Plasmodiophora brassicae* (Clubroot) Pathotype 3H under Different Resistance Genes and Initial Inoculum Levels

**DOI:** 10.3390/plants13111540

**Published:** 2024-06-02

**Authors:** Rui Wen, Tao Song, Nazmoon Naher Tonu, Coreen Franke, Gary Peng

**Affiliations:** 1Saskatoon Research and Development Centre, Agriculture and Agri-Food Canada, 107 Science Place, Saskatoon, SK S7N 0X2, Canada; rui.wen2@agr.gc.ca (R.W.); songtao@edgene.com.cn (T.S.); tonu717@yahoo.co.in (N.N.T.); 2Nutrien Ag Solutions, 201-407 Downey Road, Saskatoon, SK S7N 4L8, Canada; coreen.franke@nutrien.com

**Keywords:** *Brassica napus*, canola, resistance, gene pyramiding, *Plasmodiophora brassicae*, pathotype, resting spores

## Abstract

In this study, we explored the resilience of a clubroot resistance (CR) stacking model against a field population of *Plasmodiophora brassicae* pathotype 3H. This contrasts with our earlier work, where stacking CRaM and Crr1rutb proved only moderately resistant to pathotype X. Canola varieties carrying *Rcr1*/*Crr1^rutb^* and *Rcr1* + *Crr1^rutb^* were repeatedly exposed to 3H at low (1 × 10^4^/g soil) and high (1 × 10^7^/g soil) initial resting spore concentrations over five planting cycles under controlled environments to mimic intensive canola production. Initially, all resistant varieties showed strong resistance. However, there was a gradual decline in resistance over time for varieties carrying only a single CR gene, particularly with *Crr1^rutb^* alone and at the high inoculum level, where the disease severity index (DSI) increased from 9% to 39% over five planting cycles. This suggests the presence of virulent pathotypes at initially low levels in the 3H inoculum. In contrast, the variety with stacked CR genes remained resilient, with DSI staying below 3% throughout, even at the high inoculum level. Furthermore, the use of resistant varieties, carrying either a single or stacked CR genes, reduced the total resting spore numbers in soil over time, while the inoculum level either increased or remained high in soils where susceptible Westar was continuously grown. Our study demonstrates greater resistance resilience for stacking *Rcr1* and *Crr1^rutb^* against the field population of 3H. Additionally, the results suggest that resistance may persist even longer in fields with lower levels of inoculum, highlighting the value of extended crop rotation (reducing inoculum) alongside strategic CR-gene deployment to maximize resistance resilience.

## 1. Introduction

Clubroot, caused by *Plasmodiophora brassicae* Woronin, poses a significant threat to canola (*Brassica napus* L.) production in Canada, potentially leading to yield losses of up to 100% in severe cases [1]. The disease is also common on other brassica crops globally [2]. Motile zoospores of *P*. *brassicae*, released from germinating resting spores, typically infect root hairs, where secondary zoospores are generated through the cleavage of plasmodia. Upon release into the soil, these zoospores are able to invade the main roots and subsequently infect the cortical tissue [3], resulting in hypertrophy of root tissues and the formation of characteristic club-shaped galls. Above-ground symptoms, such as restricted growth, wilting, and premature ripening, may also occur due to reduced water and nutrient uptake from clubbed roots [4]. After swathing, these galls begin to break down in the soil, and plasmodia undergo cleavage, releasing resting spores back into the soil. While some of these spores remain viable inoculum for years [5], the majority may disintegrate fairly quickly under field conditions [6,7].

Genetic resistance has emerged as a cornerstone for clubroot management, with 11 resistance genes identified in the A genome of brassicas, including *Crr1*, *Crr2*, *Crr4*, *CRk*, *CRc*, *CRa*, *CRb*, *Rcr1*, *Rcr8* and *Rcr9* on chromosomes A01, A02, A03, A06 and A08 [8,9,10,11,12,13,14,15,16]. Although *CRa*, *Rcr1* and *Rcr4* have been shown to be identical recently [17], there are additional clubroot resistance (CR) genes that can be tapped from the A genome. In Canada, the initial resistant canola cultivar 45H29 was introduced in 2009, featuring an A03 CR gene from the European winter rapeseed cv. Mendel [18]. Subsequently, more resistant cultivars became available alongside 45H29, and together these cultivars were regarded as ‘first-generation’ resistance, carrying a single CR gene. These cultivars were resistant to five pathotypes identified initially using the common Williams’ system, but they might also impose strong selection pressure against a genetically diverse pathogen population in Canada where more than 40 pathotypes have been reported [19,20]. Resistance erosion was observed in many canola fields only 3–4 years after the introduction of first-generation resistance.

To address this challenge, it is useful to develop new CR forms targeting newly identified virulent pathotypes for improved resistance performance. Previous studies have shown that canola possessing single-gene resistance is unlikely to last when exposed repeatedly to the same source of *P. brassicae* inoculum [21]. Combining CR genes with differential resistance against *P. brassicae* pathotypes can increase the spectrum, as well as the durability of resistance, for continuously evolving pathogen populations [22]. When used with extended crop rotation, the CR-gene stacking may further deter inoculum buildup in soil, enhancing efficacy and resilience of resistant cultivars [6,23]. As the CR gene from Mendel (A03), identical to *CRa*, *Rcr1* and *Rcr4* [17], is present in the majority of the first-generation resistant canola cultivars [18], additional genes from A08 that demonstrate efficacy to novel pathotypes [16,24] may be promising candidates for gene stacking. New cultivars with stacked A03 and A08 CR genes are known as part of the ‘second generation’ resistance in Canada; this CR-gene stacking has demonstrated improved resistance efficacy, as well as resilience, against a field population of *P. brassicae* pathotype X, relative to any of the single genes alone [25]. Chinese cabbages carrying the CR genes *CRa*, *CRk* and *CRc* showed much improved resistance, relative to any of the single gene alone, against six field isolates of *P. brassicae* [14]. This demonstrates the value of CR-gene stacking against diverse clubroot pathogen populations.

In western Canada, pathotype 3H of *P. brassicae* is the most prevalent [19]. While the A03 or A08 CR gene alone or in combination can be highly effective against this pathotype, field populations of this pathogen often consist of multiple pathotypes, including potentially virulent strains at low levels. Moreover, the composition of pathogen populations can change over time [24]. It remained unclear whether stacking CR genes would provide more durable resistance against pathotype 3H compared to using either gene alone. Additionally, there was interest in understanding whether the resistance would last longer under conditions of lower soil inoculum, as extended crop rotation can effectively reduce soil inoculum levels [6], which may complement cultivar resistance. The objectives of this study were to evaluate the resilience of canola varieties carrying single versus stacked CR genes (A03 and A08) in terms of resistance efficacy and durability against pathotype 3H under conditions mimicking low and high soil inoculum levels during continuous cropping of canola. Furthermore, the study also examined the impact of these canola varieties on soil inoculum buildup over time to gain insights into the potential benefits of extended crop rotation to CR deployment for sustainable clubroot management.

## 2. Materials and Methods

### 2.1. Canola Plants 

Three inbred/hybrid canola varieties, designated as CPS#13, CPS#14, and CPS#20, were used for this study. These varieties were provided by Nutrien Ag Solutions (Saskatoon, SK). CPS#13 and CPS#20 are inbreds carrying the CR gene *Rcr1* and *Crr1^rutb^* on chromosomes A03 and A08, respectively, while CPS#20 is a hybrid of these inbreds that carries both *Rcr1* and *Crr1^rutb^* [25]. Furthermore, the susceptible variety Westar was included as a control to confirm conducive infection conditions in each trial cycle and for comparisons against these resistant varieties. Detailed information regarding these canola varieties is provided in Table 1.

For the assessment of resistance durability, seeds were planted in Sunshine #3 potting mix (SunGro Horticulture, Vancouver, BC, Canada) in 8-inch pots (6 inches deep). In each planting cycle, each pot contained 23 plants of a test variety with three replicates (pots). Seeded pots were positioned in saucers and placed in a growth room set at 22/16 °C (day/night) with a 16 h photoperiod until root sample collection. Plants were watered daily by filling the saucers, allowing water to be absorbed from the bottom. Yellow or dying leaves were removed twice weekly throughout the experiment to minimize infestation by insect pests and foliar fungal pathogens.

### 2.2. The Pathogen Inoculum

The inoculum, primarily consisting of *P. brassicae* pathotype 3H, originated from root galls gathered from commercial canola fields near Leduc, Alberta. This inoculum was maintained by continuous inoculation of the susceptible canola variety Westar under controlled-environment conditions. For inoculum preparation, dried root galls stored in a −20 °C freezer were blended with deionized water at high speed for approximately 1 min. The resulting homogenate was then filtered through eight layers of cheesecloth to yield a resting spore suspension. The concentration of this suspension was estimated using a haemocytometer and adjusted to 1 × 10^7^ spores/mL to be used to infest the potting mix (soil) in the initial cycle of exposure for the assessment of resistance durability.

### 2.3. Experiments for Clubroot Resistance Durability

Before the initial planting, the *P. brassicae* resting spore suspension was blended with soil to achieve concentrations of approximately 1 × 10^4^ and 1 × 10^7^ spores/g soil, respectively. This was designed to simulate low and high initial soil inoculum levels, resembling lightly and heavily infested field soils. In subsequent planting cycles, soil from the previous cycle was reused within the same replicate (pot), with all roots, whether with or without galls, being buried in the soil to simulate the reintroduction of inoculum from diseased plants. The soil mixture was covered with aluminum foil and left at room temperature for four weeks to facilitate the decomposition of galls and the post maturation of resting spores. Throughout this process, the soil mixture was kept lightly moist by adding 200 mL of water at the beginning and two weeks into the process. Although some of the resting spores from the initial inoculation may perish over time (Peng et al., 2015) [6], root galls from the previous planting cycle could replenish some of the soil inoculum for the following cycle, with the inoculum level being contingent upon the disease severity (gall sizes) from the previous cycle. Each experiment comprised a total of five planting cycles, exposing these resistant varieties repeatedly to the same source of inoculum, with each cycle taking about 10 weeks from planting to preparing the infested soil for the subsequent cycle. The experiment was repeated once, with all plant materials and pathogen inoculum being prepared separately for each trial repetition.

### 2.4. Disease Assessment

For each planting cycle, the roots of all plants were removed and separated at about six weeks post planting, then examined for galling with a 0–3 scale described initially by Kuginuki et al. [26], where 0 = no gall formation and 3 = severe galling on the main root. A disease severity index (DSI) was calculated for each replicate using the following formula:$$\mathrm{DSI}\frac{\Sigma \left(\mathrm{rating}\;\mathrm{class}\right)\times \left(\mathrm{number}\;\mathrm{of}\;\mathrm{plants}\;\mathrm{in}\;\mathrm{the}\;\mathrm{rating}\;\mathrm{class}\right)}{\mathrm{Total}\;\mathrm{number}\;\mathrm{of}\;\mathrm{plants}\times 3}\times 100$$

### 2.5. Quantification of P. brassicae Inoculum in Soil

Prior to the planting for each cycle, the *P. brassicae* resting spores in the soil mixture were quantified using Droplet Digital PCR (ddPCR) following a protocol developed earlier (Wen et al., 2019). Approximately 2 g of soil were sampled from five random positions/depths of each replicate before planting. After air-drying for two weeks in a Petri dish, 0.1 g of each soil sample was placed in PowerBead tubes in a DNA extraction kit (Qiagen, Toronto, ON, Canada). The soil samples were homogenized using a FastPrep 24 Homogenizer (MP Biomedical, Solon, OH, USA) for cell lysis of resting spores [27]. The remaining steps of DNA extraction were carried out following the instructions of the kit. All DNA samples were eluted in 200-µL elution buffer provided by the manufacturer and stored at −20 °C until use.

The quantification of resting spores in each soil sample was performed on a QX200TM ddPCR System (Bio-Rad, Montreal, QC, Canada); all reagents and supplies, including droplet generator oil, DG8^TM^ cartridges and gaskets, droplet reader oil, and ddPCR supermix for probe, were purchased from Bio-Rad. The primers and probe were the same as previously described [27]. Briefly, the ddPCR process began by partitioning the reaction mix containing the probe supermax, primers, probe and sample DNA into aqueous droplets in oil via the droplet generator. After transferring the droplets to a 96-well PCR plate, a thermocycling process that included incubation at 95 °C for 10 min; 94 °C for 30 s, 40 cycles; 60 °C for 60 s, 40 cycles (ramp rate set to 2 °C/s); 98 °C for 10 min and 4 °C infinite was carried out in a conventional thermal cycler. The PCR plate was then transferred to the droplet reader for automatic assessment of samples. QuantaSoft^TM^ Software version 1.7 was used for data analysis. To differentiate positive and negative droplets, the threshold was set at 2000 based on a standard operating procedure developed in house. 

### 2.6. Data Analysis

All statistical analysis was performed using the SAS version 9.3 (SAS Institute, Cary, NC, USA). DSI data were used to statistically analyze disease severity levels. The homogeneity of variance was checked with Levene’s Test prior to pooling the DSI data from different repetitions for analysis of variance (ANOVA). Fisher’s Protected LSD was used to separate the means only when ANOVA was significant (*p* ≤ 0.05). The data for ddPCR quantification of resting spores in soil samples were Log10 transformed prior to analysis, and the transformed data showed a normal distribution based on the Shapiro–Wilk Test. Resting spore concentrations were analyzed also using ANOVA and protected LSD, followed by regression analysis for the trend of soil inoculum associated with each canola variety grown continuously over five planting cycles.

## 3. Results

### 3.1. Resistance Performance of Canola Varieties 

Compared to the susceptible Westar (control), each of the resistant varieties, whether carrying a single (*Rcr1* or *Crr1^rutb^*) or stacked CR genes (*Rcr1* and *Crr1^rutb^*), exhibited high levels of resistance in the initial cycle of exposure at both low (10^4^ spores/g soil) and high (10^7^ spores/g) inoculum levels (Figure 1). Only mild disease symptoms were observed on CPS#20 (*Crr1^rutb^*) at the higher initial inoculum level, with the averaged DSI being consistently <10% (Figure 1B). Over time, however, with the continuous reintroduction of diseased roots into the soil for subsequent planting cycles, clubroot remained severe on Westar, with DSI increasing from 70% to 80% under the lower initial inoculum conditions and generally remaining above 80% for the high inoculum within the span of five planting cycles (Figure 1A,B). 

For resistant varieties, there was an tendency of increased disease over time, particularly for those carrying only a single CR gene and under high initial inoculum conditions, most notably CPS#20, which carries *Crr1^rutb^* (Appendix A), where DSI reached 38.7% in the fifth cycle (Figure 1B). In contrast, the variety with stacked CR genes (CPS#14) exhibited the strongest resistance resilience; clubroot symptoms were not observed until the fourth planting cycle even under high inoculum levels, with only minor galling on a few plants (Appendix A), resulting in an average DSI consistently below 3% in all planting cycles (Figure 1A,B). 

### 3.2. Effect on Soil Inoculum Buildup 

The initial inoculum levels fell within the parameters of the experimental design, specifically 10^4^ and 10^7^ spores/g soil for low and high resting spore concentrations, respectively, at the outset. As susceptible Westar was continuously planted and root galls recycled back into the soil, the resting spore concentration generally increased over time in both low and high initial soil inoculum cases, although the initial rise was more rapid with the low initial inoculum level (Figure 2). However, after the third planting cycle, the soil inoculum buildup noticeably slowed down, relative to the earlier cycles. By the onset of the fifth cycle, the soil inoculum had reached approximately 10^8^ resting spores/g soil for both low and high initial inoculum treatments (Figure 2A,B).

In contrast, the continuous planting of resistant varieties, whether carrying a single or stacked CR genes, demonstrated a consistent reduction of resting spores in the soil under both low and high initial inoculum conditions (Figure 2). At the lower initial inoculum, the resistant varieties decreased the spore concentration by approximately 10-fold, to just below 10^4^ spores/g soil, at the beginning of the fifth cycle. With the high initial inoculum, however, the reduction was even more substantial, decreasing the resting spore concentration in the soil by almost 40-fold over the same experimental duration, although the inoculum level still remained at around 8 × 10^5^ spores/g soil. All the resistant varieties appeared to exhibit a similar effect in reducing the inoculum buildup in the soil under both low and high initial inoculum levels (Figure 2A,B).

## 4. Discussion

The diversity of the *P. brassicae* population in Canada [19] makes it a challenge likely for any CR gene alone to be resilient, as evidenced by the rapid breakdown of first-generation resistant canola cultivars. As a result, stacking of CR genes has been widely pursued in the development of second-generation resistant cultivars, including the use of *Rcr1*/*CRa^M^* and *Crr1^rutb^* [25], as the strategy can improve resistance performances and durability, as shown in rice against *Xanthomonas oryzae* pv. *oryzae* [28]. In our earlier work, stacking *CRa^M^* and *Crr1^rutb^* showed moderate resistance against a field population of pathotype X, and the efficacy appeared more durable than using either single gene alone [25]. Furthermore, the resistance based on gene stacking effectively prevented the buildup of pathogen inoculum in soil despite the moderate resistance efficacy. Pathotype X was the first novel strain of *P. brassicae* identified that overcame first-generation clubroot resistance in Canada. In contrast, pathotype 3H is the predominant pathotype on canola [19], and most CR genes used in Canadian canola cultivars, including *Rcr1*/*CRa^M^* and *Crr1^rutb^*, are resistant to it. However, there is an increasing body of evidence indicating that a field population of any pathotype can consist of multiple genotypes, including novel types, allowing eventual adaptation of the pathogen to new CR genes deployed [29]; the higher the soil inoculum level, the greater the likelihood for novel pathotypes to be present at a level that results in visible resistance erosion [30]. The current study aimed to determine whether stacking CR genes and/or reducing the soil inoculum level would help extend the resistance durability against a field population of pathotype 3H. 

The gradual increase in DSI observed on these resistant varieties over time, particularly those carrying only *Rcr1* or *Crr1^rutb^* individually (Figure 1), suggests the presence of novel pathotypes virulent to these CR genes within this field population of 3H, consistent with previous findings in field studies [31]. Additionally, the result suggests that the virulent inoculum likely increased in the soil over time. In contrast, the variety carrying both *Rcr1* and *Crr1^rutb^* exhibited a delayed onset of disease and less pronounced increases in DSI over time, especially under lower initial inoculum conditions. The specific identities of these virulent pathotypes within this population of 3H remain unclear; however, individually, *Crr1^rutb^* appeared to be less effective than *Rcr1* against them, as evidenced by its earlier onset of disease and more rapid increase in DSI over time. This may indicate differential reactions of the two CR genes towards these virulent pathotypes, while their combination resulted in enhanced efficacy, as shown by the resistance of CPS#14 (*Rcr1* + *Crr1^rutb^*). These results underscore the potential of stacking CR genes to achieve prolonged resistance performance despite the efficacy of individual genes observed initially. The resilience of resistance is likely dependent on the composition and inoculum level of novel pathotypes present in the background. This study highlights that reducing the resting spore concentration from 3 × 10^7^ to 10^5^ spores/g soil (measured numbers) can not only enhance the effectiveness of resistant varieties but also contribute to their durability by maintaining relatively low disease levels regardless of whether single or stacked CR genes are involved. However, stacked genes consistently demonstrate slight advantages in terms of greater resistance efficacy and durability compared to either single gene alone.

Continuous planting of resistant varieties led to substantial declines in total resting spore numbers in the soil, regardless of initial inoculum levels. In contrast, spore numbers increased in the Westar controls. The decline was more pronounced with higher initial inoculum treatments, showing approximately a 40-fold reduction over four planting cycles compared to a 10-fold reduction with lower initial inoculum levels over the same duration. This trend remains relatively consistent for each resistant variety regardless of whether single or stacked CR genes are involved. Since these CR genes, either individually or in combination, are highly effective against pathotype 3H, their initial efficacy was significant, especially as the inoculum primarily consists of 3H. Consequently, the total resting spore concentration would be drastically reduced, even though the virulent inoculum from infected roots, with or without clubroot galls, was recycled back into the soil. It is worth noting that the reduction in soil inoculum by the variety carrying stacked CR genes was not substantially different from that of single-gene varieties (*p* > 0.05, LSD) despite its demonstrated advantages in resistance performances, such as a later onset of disease symptoms and lower DSI (Figure 2). This similarity is likely due to the relatively small portion of virulent inoculum in the soil, which is overshadowed by a much larger portion of 3H inoculum.

The resting spore concentration in the soil increased rapidly where the susceptible Westar was grown, a result of the severe DSI observed from the outset and the recycling of galls into the soil for subsequent planting. Even at the lower initial inoculum level of 10^5^ spores/g soil (measured number), the DSI was approximately 70% in the first planting (Figure 1). However, despite continuous recycling of diseased roots back into the soil, the inoculum level plateaued at about 10^8^ spores/g soil. This is likely due to a natural decline of the inoculum caused by the decomposition of immature resting spores [32], which are typically less effective for infection and often short-lived in soil [6,7]. In contrast, each of the resistant varieties resulted in a 10- to 40-fold decline in soil resting spore concentration under low and high initial inoculum conditions, respectively, while also mostly controlling the disease throughout the duration of the experiment. This highlights that the use of resistant varieties can significantly reduce soil inoculum buildup compared to the intensive cropping of susceptible canola, as demonstrated in other studies [33,34]. However, this effect on soil inoculum buildup may only apply to pathotype 3H or other avirulent pathotypes, while the inoculum of virulent pathotypes likely increased, as evidenced by the DSI increases over time, especially on the variety carrying only *Crr1^rutb^* grown under high initial inoculum conditions. This also illustrates the pattern of resistance erosion when the same CR genes are repeatedly utilized.

From a practical perspective, the results demonstrate that canola cultivars carrying either *Rcr1* or *Crr1^rutb^* alone or both CR genes in combination can be highly effective in fields predominantly affected by pathotype 3H. However, virulent pathotypes present at low levels in the background will eventually overcome this resistance. Stacking the two CR genes in a cultivar may slow down the buildup of virulence inoculum to some extent, but, as indicated by the gradual increases in DSI over time (Figure 1), the surge of virulent pathotypes cannot be completely halted by resistant canola cultivars alone. Furthermore, resistance tends to be stronger and more resilient under lower soil inoculum levels, particularly for varieties carrying stacked CR genes. This emphasizes the importance of implementing extended crop rotations in clubroot management, where a break of more than two years from a previous canola crop can reduce soil inoculum by up to 90%, relative to shorter rotations [6,33,35]. In the present study, resistant varieties steadily reduced the total number of resting spores to approximately 10^4^ spores/g soil under the low initial inoculum; although this level of inoculum can still infect susceptible canola [36], the pressure on resistant varieties is likely lighter, as demonstrated by the current study. The challenge, however, arises in heavily infested fields or patches where millions to billions of spores may be present. In such scenarios, there is a greater proportion of virulent pathotypes existing in the background, leading to rapid resistance erosion. Therefore, reducing the inoculum level should be the priority for these fields before introducing a resistant canola cultivar, preferably one carrying stacked CR genes.

## 5. Conclusions

The theory behind the longevity of resistance with R-gene pyramiding is based on the hypothesis that the probability for mutation to multiple virulence is low in the pathogen population [37]. This study examined the resilience of a CR stacking model with a high level of resistance against pathotype 3H, contrasting with our earlier research where stacking *CRa^M^* and Crr1rutb proved only moderately effective against pathotype X. While stacking *Rcr1/CRa^M^* and *Crr1^rutb^* generally enhanced CR performance in both models compared to using either single gene alone, the resistance resilience may hinge on the presence and proportion of virulent pathotypes in the background of the inoculum. The study demonstrated that using a resistant canola variety reduced the soil inoculum buildup, likely due to fewer and smaller galls returning to the soil and the natural decline in the existing soil inoculum over time. However, the proportion of virulent inoculum likely increased due to selection pressure, as evidenced by the gradually increased DSI on varieties carrying either *CRa^M^* or *Crr1^rutb^*. This highlights the potential for resistance erosion if these varieties are repeatedly used, especially under high inoculum conditions. Using varieties with stacked CR genes may decelerate resistance erosion, resulting in later onset of disease and lower DSI following multiple exposures, as demonstrated in this study. Moreover, resistance can be further prolonged under lower inoculum conditions, and extending crop rotation to reduce soil inoculum levels should be considered alongside CR deployment to maximize resistance resilience.

## Figures and Tables

**Figure 1 plants-13-01540-f001:**
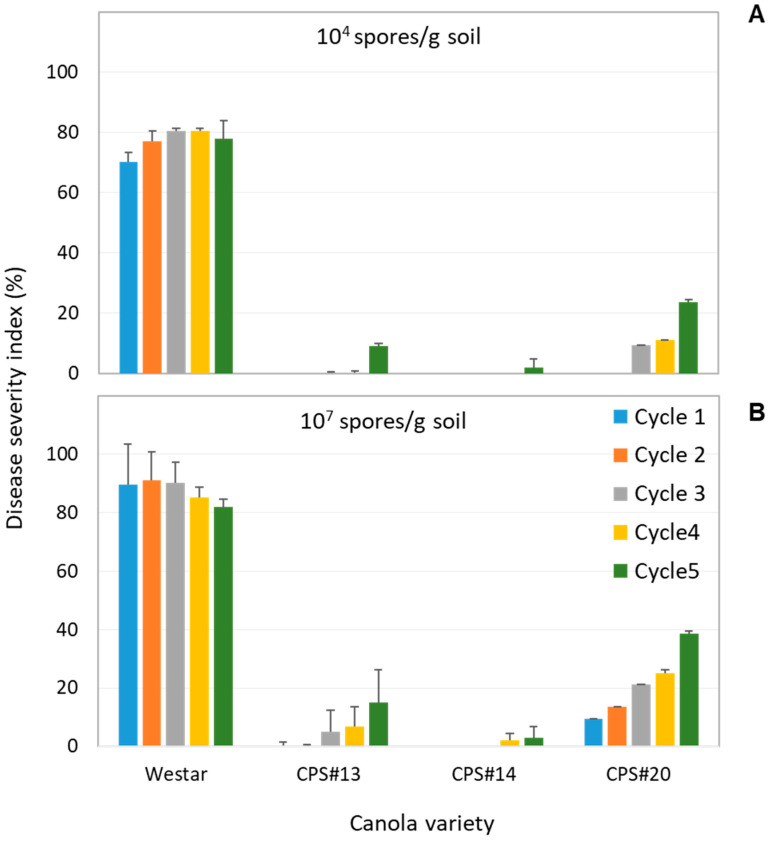
The disease severity index (DSI) on Westar (susceptible), CPS#13, CPS#14 and CPS#20 (resistant) over five planting cycles under low (**A**) and high (**B**) initial inoculum levels of P. brassicae pathotype 3H. Error bars represent standard deviation (STDEV) of DSI assessment.

**Figure 2 plants-13-01540-f002:**
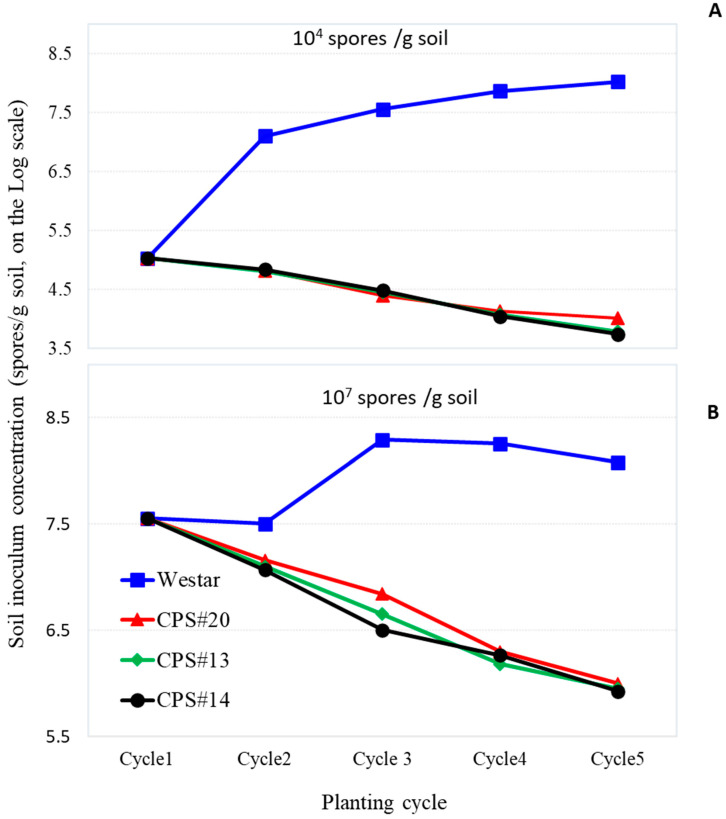
Resting spores concentrations in soils associated with Westar (susceptible), CPS#13, CPS#14 and CPS#20 (resistant) over five planting cycles. The pathogen inoculum was added at low (**A**) and high (**B**) levels, respectively, only in the initial planting cycle, and diseased roots were expected to provide additional inoculum in subsequent cycles. The level of soil inoculum was quantified for each replicate using ddPCR [27] prior to planting. Each data point was an average of six replicates (*n* = 6) from two repeated experiments. STDEV for Westar ranged from 4.4 to 7.8 and 6.8 to 8.1 on the log scale for the low and high initial inoculum levels, respectively. For the resistant varieties, the corresponding STDEV values ranged from 3.4 to 4.6 and 5.2 to 6.8.

**Table 1 plants-13-01540-t001:** Canola Inbreds/hybrids carrying one or two CR genes used for the assessment of resistance resilience against pathotype 3H of *P. brassicae*.

Variety	Internal Coding by the Breeding Company	CR Genes Involved	# CR Genes
CPS#20	SC15-NB3-01	*Crr1^rutb^*/*Crr1^rutb^*	ONE
CPS#13	PS-FCA 15-3978	*Rcr1*/*Rcr1*	ONE
CPS#14	PS-ARK 14-3562	*Rcr1*/*Crr1^rutb^*	TWO
Westar	N/A ^+^	None	ZERO

^+^ N/A: Not applicable.

## Data Availability

All datasets from this study are included in the article.

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
