# Peer review of "Resilience of Canola to *Plasmodiophora brassicae* (Clubroot) Pathotype 3H under Different Resistance Genes and Initial Inoculum Levels"

_plants, 2024, doi:10.3390/plants13111540_

Round 1
Reviewer 1 Report
Comments and Suggestions for Authors
This study tests long-term clubroot resistance of three canola cultivars with single-gene vs. two-gene resistance in low-inoculum and high-inoculum conditions. This is a straightforward, well-designed experiment with clear results that lead to practical recommendations for canola breeders and producers. It is well-written, and the figures are clear and well-described.
Minor text edits:
Line 56: “taped” should be “tapped”.
Line 82: Change “gene” to “genes”.
Line 107: CPS #20 should be CPS #14.
Line 186: “statistical analysis” should be “statistically analyze”
Line 254: Change “either the single gene alone” to “either single gene alone”.
Line 264: Change “result” to “results”.
Author Response
Thank you for your careful review of the manuscript. All suggestions have been adopted, except for the one regarding "Line 107: CPS #20 should be CPS #14." CPS #20 correctly represents a single-gene variety, as opposed to the double-gene variety CPS#14. The numbering system may seem unintuitive for the manuscript; however, it was provided by breeders for the 20 varieties used in our initial resistance assessment, and we have not changed the numbering to avoid confusion amount different users of the information.
Reviewer 2 Report
Comments and Suggestions for Authors
The reviewed manuscript, titled "Effect of Gene Stacking and Soil Inoculum Level on Resistance Resilience of Canola Against Pathotype 3H of Plasmodiophora brassicae (Clubroot)" by Wen et al. is a fundamentally sound analysis of the effects of stacking multiple resistance genes into the agriculturally vital crop canola to the clubroot pathogen. The authors compare several strains of canola and cultivate under low versus high soil conditions of the pathogen. Overall this work is adequate and it clearly provides an answer to the question that the authors were investigating. My biggest critique/concern is simply that this is not a complete study that the heading title of an 'article' would imply that it is. It is, at best, a note/investigation/short report/etc - some small form manuscript. If the authors or the editor simply relabel or rebrand this work as that my major concerns would be alleviated. As written I would reject this work as not being a complete article.
The following comments are provided to help the authors improve this work as if it were a brief report.
Major comments:
The introduction is clear and well written. The background seemed clear and appropriate, was concise yet included sufficient detail for this reviewer to clearly follow the design and rationale for the work - even though this specific area of research is more tangential to my own. The methods section is particularly excellent. This reviewer has flagged this paper - with the intention of providing it to students as an outstanding example of how to write a methodology section involving plants and growth/culture conditions. The results are clear and the conclusions seem appropriate based on the limited experiments that were performed.
Minor comments:
Figure 1: Format better. The graphs are not aligned properly (left boundary justification specifically). The quantification of spore conc. is in the legend, not sure that it is needed in the header of each graph as well. Error bars lack explanation in the legend of the figure. I am unclear why the % error bars should go over 100% based on the normalization and severity index that the authors are using. Please clarify, revise, and edit appropriately.
Figure 2: similar comments to figure 1. In addition, there is a punctuation in the axis label (.Lg spores/g) that seems a typo. There are multiple experiments that were performed and averaged for these plots, please include error bars for St. Dev. or variance.
Author Response
Thank you for your careful review of the manuscript and your helpful comments and suggestions. Below, we provide brief responses to the more substantive concerns, explaining the changes made during the revision to address the issues raised.
1) My biggest critique/concern is simply that this is not a complete study that the heading title of an 'article' would imply that it is. It is, at best, a note/investigation/short report/etc - some small form manuscript. If the authors or the editor simply relabel or rebrand this work as that my major concerns would be alleviated. As written I would reject this work as not being a complete article.
Response: The title has been revised to better define the scope of the study. This was an extensive investigation conducted over two years of continuous greenhouse experimentation, with each repetition (five planting cycles) taking roughly 15 months. Each step required extreme care and precision to avoid errors that could compromise the entire experiment. The study was both time-consuming and resource-intensive. However, this information is highly needed as growers consider management strategies involving the deployment of resistant canola cultivars and extended crop rotation to reduce soil inoculum for sustainable canola production.
Overall, we believe this study provided information that addresses three critical questions:
- A resistant variety, carrying either a single or stacked CR genes, should be considered for managing clubroot caused by pathotype 3H to reduce disease impact and soil inoculum buildup.
- Varieties with stacked genes can be more efficacious than those with a single gene alone, improving resistance resilience, especially under low soil inoculum levels.
- Reducing soil inoculum through extended crop rotation is beneficial for the performance of resistant varieties.
Extensive discussion is necessary to interpret the results and draw these conclusions. Therefore, a comprehensive article will help fully convey the scope and findings of this work. We hope the revised title more accurately reflects the breadth of the study, alleviating any perception that the investigation is incomplete. Abstract and Conclusions (Lines28-34; 332-338) have been modified slightly to highlight these key findings.
2) Figure 1: Format better. The graphs are not aligned properly (left boundary justification specifically). The quantification of spore conc. is in the legend, not sure that it is needed in the header of each graph as well. Error bars lack explanation in the legend of the figure. I am unclear why the % error bars should go over 100% based on the normalization and severity index that the authors are using. Please clarify, revise, and edit appropriately.
Response: The figure has been refined for better alignment. We have saved it in a TIFF file to maintain the original settings and resolution. The legend and caption have been modified to minimize duplication. In two cases for Westar under the high inoculum level (B), error bars exceeded the 100% maximum value due to the high mean DSI and the variability of data in those treatments. This should be clear to most readers.
3) Figure 2: similar comments to figure 1. In addition, there is a punctuation in the axis label (.Lg spores/g) that seems a typo. There are multiple experiments that were performed and averaged for these plots, please include error bars for St. Dev. or variance.
Response: The figure has been refined and the typo rectified. We initially tried to include the STDEV bars, but they extended beyond the graph for Westar (susceptible). Additionally, the bars for the three resistant varieties were crowded together or overlapped. The STDEV information has now been included in the figure caption (Lines 494-497) to show the range of values associated with the susceptible and resistant varieties under low and high initial inoculum conditions, respectively.

Reviewer 3 Report
Comments and Suggestions for Authors
Thank you very much for given me the chance to read the manuscript “Effect of Gene Stacking and Soil Inoculum Level on Resistance 2 Resilience of Canola Against Pathotype 3H of 3 Plasmodiophora brassicae (Clubroot)” by Wen et al. This manuscript reported the resilience of clubroot resistance crop exposed repeatedly to the Plasmodiophora brassicae pathotype 3H. Clubroot is one of the most invasive disease for the cruciferous crop word wide, the results presented in this study can provide useful information for crop planting in the clubroot inoculated field. I suggest to accept for publication in Plants
Author Response
Thank you for reviewing the manuscript and for your positive comments. We have thoroughly reviewed the manuscript once more to ensure all information is correct and all comments/questions have been addressed. Please let us know if you have any further comments on the revision.

Round 2
Reviewer 2 Report
Comments and Suggestions for Authors
This work should be properly labeled as a brief report or a communication. It does not constitute a complete study that the label of article implies. Aside from that, the authors have made an effort to correct the formatting and omissions of key details that were needed and suggested by this reviewer.
Author Response
If it is acceptable to the editors, the labeling of the article will not be a major issue for us. It is more important to disseminate the information to canola producers and the crop industry for meaningful extension messaging.